# Deep-Learning-Based Network for Lane Following in Autonomous Vehicles

**Abida Khanum** [1]**, Chao-Yang Lee** [2,]*** and Chu-Sing Yang** [1]

1    Department of Electrical Engineering, National Cheng Kung University, No. 1, University Road,
     Tainan City 701, Taiwan
2    Department of Aeronautical Engineering, National Formosa University, No. 64, Wunhua Rd.,
     Huwei Township, Yunlin Country 632, Taiwan
*    Correspondence: chaoyang@nfu.edu.tw

**Abstract:** The research field of autonomous self-driving vehicles has recently become increasingly popular. In addition, motion-planning technology is essential for autonomous vehicles because it mitigates the prevailing on-road obstacles. Herein, a deep-learning-network-based architecture that was integrated with VGG16 and the gated recurrent unit (GRU) was applied for lane-following on roads. The normalized input image was fed to the three-layer VGG16 output layer as a pattern and the GRU output layer as the last layer. Next, the processed data were fed to the two fully connected layers, with a dropout layer added in between each layer. Afterward, to evaluate the deep-learning-network-based model, the steering angle and speed from the control task were predicted as output parameters. Experiments were conducted using the a dataset from the Udacity simulator and a real dataset. The results show that the proposed framework remarkably predicted steering angles in different directions. Furthermore, the proposed approach achieved higher mean square errors of 0.0230 and 0.0936 and and inference times of 3–4 and 3 ms. We also implemented our proposed framework on the NVIDIA Jetson embedded platform (Jetson Nano 4 GB) and compared it with the GPU's computational time. The results revealed that the embedded system took 45–46 s to execute a single epoch in order to predict the steering angle. The results show that the proposed framework generates fruitful and accurate motion planning for lane-following in autonomous driving.

**Keywords:** deep learning; gated recurrent units; VGG16; lane following; decision making

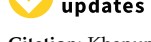



## 1. Introduction

Recently, self-driving autonomous cars and vehicles have become a new research field within the field of artificial intelligence, and this is expected to evolve soon. Self-driving technology is expected to significantly impact the automotive industry, and it ensures safe driving practices by automatically controlling vehicles. Thus, it reduces road accidents and economic damage and is a safer alternative to human drivers. An autonomous vehicle (AV) is a driverless car or one with non-human communication. Autonomous driving has six different levels: zero, little, half, conditional, high, and full automation (Figure 1). Li et al. [1] presented an active traffic lane management method for intelligent connected vehicles for optimal urban driving. The performance could be optimized to improve the traffic capacity in urban areas, lane management, and traffic congestion on freeways. Simmons et al. [2] presented a lane-following model based on a DNN and CNN; the objective was to improve accuracy and loss. The method assigned the information of the surrounding cars as the input and predicted the motion control task. Wu et al. [3] developed a model that predicted and analyzed heavy traffic conditions using LSTM. They evaluated their proposal using the publicly available NGSIM US-101 to improve the performance in terms of the root mean square error (RMSE).

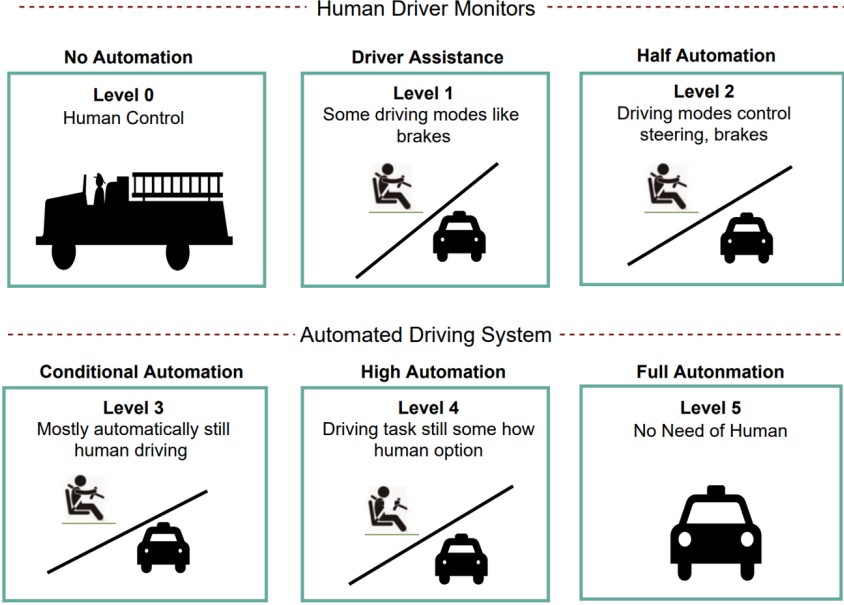

**Figure 1.** Six levels of AVs: Monitoring by a human driver (**Top**) comprising three subparts—no automation, driver assistance, and half automation—and an automated driving system (**Bottom**) comprising three subparts—conditional, high, and full automation.

Regardless of the evolution of traditional autonomous driving systems, many limitations still exist that prevent the development of full AVs. Herein, we introduce a deep-learning-based method for lane following. The proposed approach is called the VGG–gated recurrent unit (VGG-GRU). It is based on an understanding of the planning of the lane following and the control of the steering angle with speed to continuously hold the autonomous car in the middle of the path during a simulated driving situation and real scenarios. The proposed VGG-GRU does not use any integrations of different models to predict the steering angle with speed multiple controls, and it reduces the computation times of different prediction controls, which can be predicted simultaneously in both real and simulated scenarios. To calculate the improvement in the performance of VGG-GRU, the loss, root means square error (RMSE), parameters, and inference time are used.

For human-like driving, Xia et al. [4] proposed the Human-Like Lane-Changing Intention Understanding Model (HLCIUM) to understand the lane changes made by vehicles according to human perception. The proposed method was fed into the framework after preprocessing using VGG16 transfer learning to extract information, and the remaining part of the proposed method was based on the output of the VGG16 layer. The final result of the output of the VGG16 transfer learning layer was used to connect the hidden GRU layer, and the output of the GRU layer was connected with the full layer to calculate the pattern. Finally, the decision-making control comprised the steering angle and speed. Therefore, the model training and simulations were conducted by using tracks on the Udacity platform and the real dataset of Lincoln's car. An on-road stumbling block was captured by three cameras—left, center, and right—and four attributes were used: the steering angle, throttle, brake, and speed. The real dataset was captured with a single camera, and four attributes were used: the steering angle, throttle, brake, and speed. However, herein, we used raw images of environments as well as the steering angle and speed to train the proposed framework. The images were sequentially fed into three layers, with the VGG16 output layer as a pattern and the GRU output layer as the last layer. Next, the processed data were fed to two fully connected layers, with a dropout layer added between each layer. Afterward, to evaluate the deep-learning-network-based model, the steering angle and speed were predicted from the control task as the output for lane following during autonomous driving. The main purpose of the deep learning network of an AV is to build an autonomous car with good lane following and to improve

the performance metrics, such as the mean square error (MSE), root mean square error (RMSE), inference time, and number of parameters, in comparison with those of other deeper networks. To ensure the performance of autonomous driving, the computing time of the deep learning algorithm was assessed so that the model in could be implemented in real time. However, the proposed framework was run on an embedded system to test the model in real time in order to achieve better performance in terms of the runtime for a lower power consumption. Herein, we compare the runtime when using a GPU and an embedded system. The execution predicted the runtime needed to calculate a single epoch. The proposed VGG-GRU should be able to control the steering angle to maintain the lane following of the AV in order to safely drive without crashing or leaving the road (Figure 2).

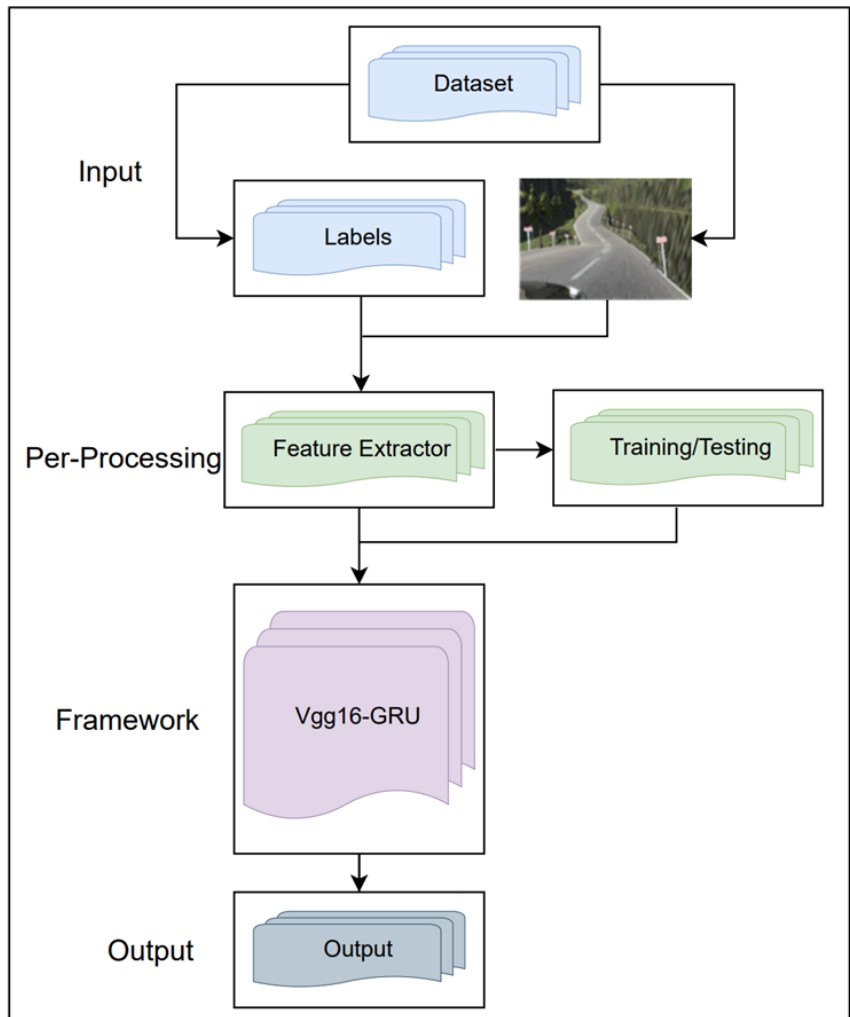

**Figure 2.** Overview of the proposed framework.

The key contributions of this article are as follows:

1. A human-like decision-making motion planning approach (i.e., lane-following) that uses the VGG16-GRU algorithm was proposed. We used a simulated driving dataset and a real dataset to test our proposed method.
2. An analysis of the result of the proposed VGG16-GRU model was carried out with two processing networks—an embedded system and a GPU—in order to analyze the power consumption.
3. We compared the VGG16-GRU framework proposed herein with other proposed networks to demonstrate its better performance.

4.   A detailed analysis of the results is provided in terms of  four performance measures: MSE, RMSE, time, and the number of parameters.

The remainder of the study is as follows: Section 2 addresses the background of the study. Next, Section 3 details the methodology, and Section 4 describes the simulation and experimental results. Finally, Section 5 concludes and presents directions for future research.

## 2. Related Work

In the field of self-driving vehicles, most existing studies focused on decision making in motion planning based on controlling the vehicle without leaving the road. However, this research reviews many algorithms in order to achieve better accuracy and reduce the time for the improvement of autonomous driving performance. Larsson et al. [5] presented a study on the prosocial analysis of multiple AVs using a predictive optimization algorithm. They evaluated the algorithm on both low- and high-quality traffic simulators, thereby improving the performance in terms of efficiency and safety. Wang et al. [6] used a feed-forward neural network and GRU for a deep-neural-network-based car-following model in order to improve the performance in terms of accuracy using the NGSIM (Next-Generation Simulation) dataset. Table 1 shows the critical ideas of existing studies, their methods, and the models that they proposed in the area of AVs.

**Table 1.** Table summarizing the existing studies in terms of their datasets, critical ideas, and methods for comparison with the proposed framework.

| Ref. | Dataset | Critical Idea | Method |
|:---:|:---:|:---:|:---:|
| [7] | Udacity | To improve control accuracy of the steering angle | CNN+LSTM+FC |
| [8] | TORCS | Multi-state model for performing higher-precision lane keeping | 3DCNN-LSTM |
| [9] | Real data | Motion prediction method for multilane turns at an intersection | LSTM-RNN |
| [10] | TuSimple | Lane detection by using multiple frames with a hybrid architecture | CNN-RNN |
| [11] | Comma.ai | AV control based on visual attention | CNN-RNN |
| [12] | Simulator | DRL method used for obstacle avoidance in an urban environment | DQN |
| [13] | NGSIM | Human-like decisions in lane change maneuvers | LSTM-CRF |
| [14] | Lyft | Deep-learning-based trajectory prediction for AVs | Resnet Model |
| Proposed | Simulator & Real | Deep-learning network for lane following with testing in an embedding system | VGG-GRU |

Sokipriala et al. [15] proposed a deep-transfer-learning method for the reduction of training time, improvement of accuracy, and estimation of the control command steering.

Their study integrated VGG16 and long short-term memory (LSTM), where VGG16 extracted features from the real-world Udacity dataset, and these were fed into the LSTM network to predict the steering angle in real time. Kortli et al. [16] approached real-time lane position perception with a CNN (convolutional neural network) as an encoder–decoder and LSTM trained on different dynamic and complex road states from a public dataset. Their results showed better performances, with a recall of 97.54% and F1-score of 97.42%. Chen et al. [17] presented motion planning with fuzzy logic for an AV using a reinforcement learning algorithm, namely, deep Q-learning. Different motion tasks (e.g., with the steering wheel and accelerator) had better performance and safety than that attained with other approaches. Hao et al. [18] presented a multimodal multi-task learning software and hardware codesign as a differentiable optimization problem to improve the performance and reduce power consumption. Azam et al. [19] proposed a neural-network-based controller using a path-tracking method for the controller (e.g., path tracking, longitudinal control, and behavioral cloning). The objective was to make valid predictions and show better scores for the performance metrics. Curiel et al. [20] proposed a method of estimating the steering angle of AVs based on a CNN architecture. They used datasets of real vehicles, which included the steering wheel, throttle, and brake in images. The results of their study revealed a low-cost and high-performance autonomous vehicle. Sumanth et al. [21] proposed a novel framework based on transfer learning with the VGG16 architecture. In their study, the sensor information was fed to VGG16 for learning and to predict the steering angle. Lee et al. [22] developed a system combining path prediction with a convolutional neural network (PP-CNN) in order to evaluate the real-time performance for a lane change. Li et al. [23] introduced an intention inference method based on LSTM and GRU with the aim of achieving a better inference than those attained with existing approaches. Zhou et al. [24] proposed a multitask framework for detecting objects and lanes using an embedded system to improve the runtime performance in autonomous driving. Hu et al. [25] presented a deep cascaded neural network method that integrated VGG and long short-term memory to predict multiple control tasks, such as steering, braking, and accelerating.

## 3. Methodology

This section presents the architecture, which integrates three different patterns—namely, the VGG16, the GRU, and fully connected layers—to predict the steering angle by using deep learning methods. Initially, different networks were considered, including VGG16 and the GRU. For the feature extraction, we used VGG16, which is shown in Figure 3b, as the initial method, the output of which is fed to the GRU to extract further information from the image data. Then, the last fully connected layer outputs the predicted steering angle (e.g., $L_d$, $C_d$, $R_d$) with the speed for the motion planning control tasks. Yadav et al. [26] introduced a pretraining method in order to predict the steering angle and achieved better results. Jiang et al. [27] introduced an end-to-end learning-network-based model with the VGG16 for transfer learning, which was integrated with the LSTM framework to extract information for predicting the output. The model showed better performance in terms of accuracy in less training time. Anwar et al. [28] focused on a transfer-learning-based method to reduce the computational training size of a deep neural network for autonomous navigation. Zheng et al. [29] demonstrated that a transferable learning feature reduces the global position separation between different networks by using a deep neural network (DNN), such as Faster R-CNN or YOLOV2.

A gated recurrent unit (GRU) is a type of recurrent neural network (RNN) that was presented by Cho et al. [30] in 2014. It consists of two gates, as shown in Figure 3c: the reset gate $R_t$ and the update $Z_t$ units; there is a state candidate hidden layer, and the current cell is denoted by $\widetilde{H_t}$. The equations of the GRU for controlling the mechanism of each gate are as follows:

$$R_t = \sigma(W_r s_{t-1} + U_r x_t + b_r). \tag{1}$$

$$Z_t = \sigma(W_z s_{t-1} + U_z x_t + b_z). \tag{2}$$

$$\widetilde{H_t} = \Phi(W(r_t \circ s_{t-1}) + W_{xt} + b). \tag{3}$$

$$H_t = Z_t \circ s_{t-1} + (1 - z_t) \circ \widetilde{H_t}. \tag{4}$$

where $W_z$, $W_r$, and $W$ denote the weight matrices and $U_z$, $U_r$, and $U$ are the previous time's weight matrices. In addition, $b_z$, $b_r$, and $b$ are biases and $\sigma$ denotes the sigmoid.

### 3.1. Overview of the Proposed System

The proposed framework was used in a self-driving vehicle system for lane-following decision making in order to control the steering direction (e.g., left turn (LT), right turn (RT), and lane keeping (Lk)) with speed. The overall framework is divided into six parts: the input, transfer layer (VGG), dense layer, GRU, fully connected layer, and control direction (Figure 3a).

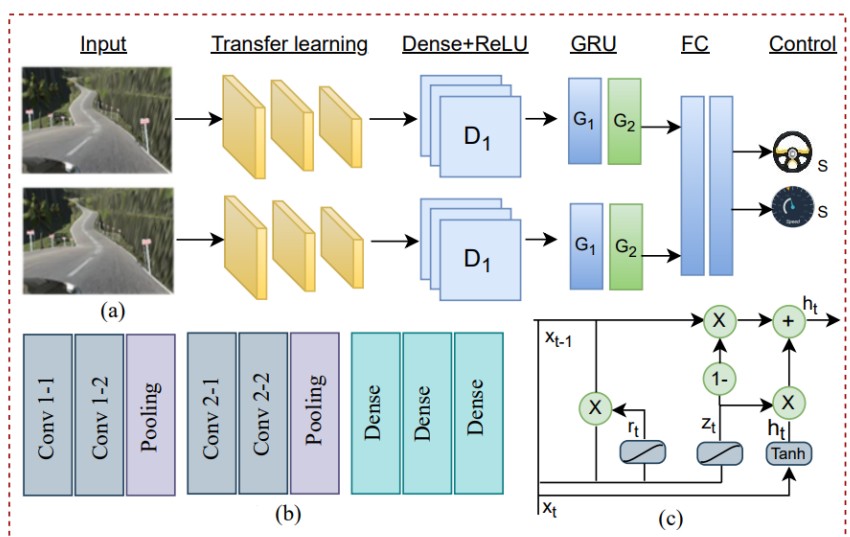

**Figure 3.** Overall proposed framework: (**a**) three parts: input, the proposed model (VGG-GRU), and the control of the steering angle and speed. (**b**) The overview of the simple VGG16 (GRU). (**c**) The structure of the gated recurrent units (GRUs).

### 3.1.1. Input Layer

Herein, Udacity and real self-driving vehicles were used to gather the datasets. The Udacity dataset was captured with three directional cameras (left, center, and right) with four variables, namely, the steering, throttle, brake, and speed. The Udacity dataset contained the data for both tracks. The Lincoln self-driving car dataset was captured from a single-direction camera with four variables, namely, steering, throttle, brake, and speed. The samples were split with a ratio of 80:20 (Table 2). The actual resolution of the images was $160 \times 260 \times 3$ dpi. The stored datasets were processed before training the proposed framework. During the preprocessing step, the original images were cropped to remove extra parts, such as trees, the sea, and the sky. However, the resized input shape was $80 \times 80 \times 3$, and this was used as the input for the framework. RGB is not the best mapping for visual perception. However, the YUV color space is better for coding and reduces the bandwidth in comparison with RGB, as shown in Figure 4, which shows both the Lack dataset and the Mountain dataset. The real image format was in BGR, so we converted them into RGB and cropped the extra parts. The resized shape was $80 \times 80 \times 3$, and this was used as the input for the proposed model.

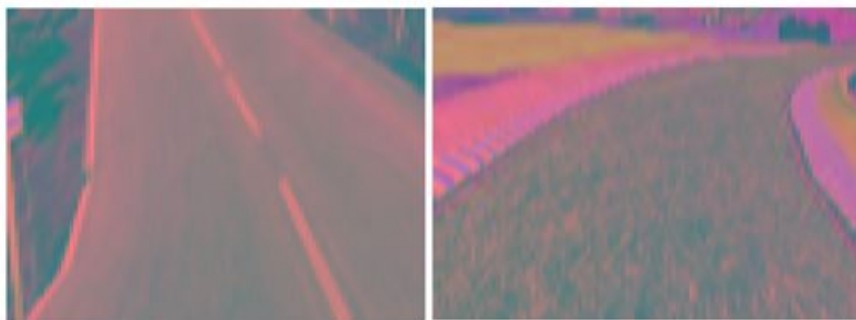

**Figure 4.** The YUV format of images from two datasets: the Mountain dataset (**left**) and the Lack dataset (**right**).

**Table 2.** Descriptions of the Udacity dataset and the real dataset.

| Data | Samples | Training | Testing | Size |
|---|---|---|---|---|
| Lack | 25,162 | 20,129 | 5033 | 1.2 GB |
| Mountain | 18,366 | 14,692 | 3674 | 512.4 MB |
| Real | 10,011 | 8008 | 2003 | 7.8 GB |
| Spilt | 100 | 80 | 20 | - |

Every image was matched with the steering angle, which was normalized in the range of $-1.5$ to $1.5$, where $-1.5$, $1.5$, and $0$ fell to the left, right, and center, respectively; these are expressed in Equation (5) for the steering angle:

$$Direction = \begin{cases} Left\ angle, & if\ x < 0 \\ center\ angle, & if\ x = 0 \\ Right\ angle, & if\ x > 0. \end{cases} \tag{5}$$

Herein, the trained model used the raw image input to predict the output with a VGG-GRU network to keep a lane-following AV on the road. The training dataset contained the input images and the labeled output of the steering direction to keep the vehicle on its path. The image data were fed into the framework after preprocessing with the method for extracting information, which is denoted by $I$. The remainder of the proposed method is based on the VGG16-GRU and the control of the steering direction. The input layer $I$ is expressed in Equation (6).

$$I_v = I_1, I_2, I_3, \cdots, I_n \tag{6}$$

3.1.2. Proposed System

A diagram of the proposed system for controlling the steering angle and speed of autonomous driving was planned. The VGG16 architecture comprised two layers, with conv use, 33 kernels, and 22 strides. The depth of the first layer was $24, 32, 48$, and then the last convolutional layers comprised a $3 \times 3$ kernel, and $1 \times 1$ strides each. The depth of each last layer was 64 or 128. Afterward, it was integrated with two GRU layers, and a fully connected was the last layer of the framework. The output was the control of tasks such as the steering angle and speed.

The image data were fed into framework after preprocessing with VGG16 transfer learning for the extraction of information, which is denoted by $I_v$, and the remaining part of the proposed method was based on the output of the VGG16 layer, which is denoted by $V_o$ and is expressed in Equation (7).

$$V_0 = V_1, V_2, V_3 \cdots V_n. \tag{7}$$

The features extracted from the previous continuous input, which are represented by $V_o$ and $l_{(g)}$, were used. Then, the last output of the network (represented by $g_{(o)}$) was

connected to the *FC* layer (represented by $l_{(o)}$) and then to the output layer ($0_t$); the outputs of these layers were the steering angle and speed.

### 3.1.3. Output

The final output ($FC$) to $D = d_1, d_2, d_3 \cdots d_n$ was used to compute the control tasks of the pattern. The outputs were treated as the decision to keep the vehicle in its lane. However, the particulars of each GRU included the individual input gate, reset gate, update gate, and output gate, The sequence layer contained two GRUs that connected the steering angle and speed with the FC layers. Next, the control tasks were predicted. The process of the proposed architecture is expressed as:

$$D_{(output)} : d(LT_s, RT_s, S) \tag{8}$$

where $D$ represents the final outputs of the proposed architecture.

With the VGG-GRU framework, the learning method for the steering angle and speed for the prediction of decisions is summarized in Algorithm 1.

---

**Algorithm 1:** Steering direction with speed control with VGG-GRU

---

*Input:* $I = I_1, I_2, I_3, \cdots, I_n$
*Output:* $D_o = LT_s, RT_s, S$

*Start ()*
*Define parameters of Vgg-GRU;*
*Given direction $LT_s, RT_s$ and speed*
*Spilt data into Training and Test $TR_d, TE_d$*
*For I in $TR_d$*
*$TR_d, TE_d$ VGG-GRU with I*
*Fit data into VGG-GRU network*
*VGG-GRU (Load-framework)*

*Execute*
*compute the function $\leftarrow$ VGG-GRU*
*MAE $\leftarrow$ VGG-GRU*
*Runtime $\leftarrow$ VGG-GRU*
*Accuracy $\leftarrow$ VGG-GRU*
*End*

---

## 4. Results

Here, the results of the presented VGG16-GRU framework were analyzed on an Intel Core i5-4440 CPU with 16 GB of RAM, Jetson Nano, and an NVIDIA GTX 2080Ti. The experiment was executed on a Jupyter notebook in TensorFlow, TensorFlow-GPU, Anacondaand Keras. For the analysis of the proposed framework, some metrics were used to evaluate the performance of the models, including the MSE, RMSE, number of parameters, and runtime. The execution runtime was computed with a single epoch and the total running time. However, we also compared the runtime and power of the proposed framework on a GPU and Jetson Nano to find the runtime for a prediction in a single epoch.

### 4.1. Simulation Framework

This section presents the experiment conducted with the proposed model. A self-driving environment was used to examine an autonomous self-driving car at a normal speed. The Udacity simulator [31] is an open-source driving environment platform used for self-driving vehicles. The graphics settings can be changed according to the user: a screen resolution of $1600 \times 1200$ was chosen and graphics quality was set to "fantastic" This platform has two modes and two simulator tracks for driving a car in an environment: the training mode and the autonomous mode. Screenshots of both tracks (the Lack track

and the Mountain track) are shown in Figure 5a,d. The Lack track is a simple, not-too-curvy track, and it is easier for cars to drive on it. The Mountain track is more complex and has sharper turns than in a village track. We tested our proposed model on both tracks. The training mode gave an option to drive the vehicle in the simulator, and the dataset was recorded, as shown in Figure 5b,e. The simulator showed a red circular sign at the top right of the screen when recording the dataset, and a folder was created, which contained an image folder and a CSV file. An on-road stumbling block was captured by three cameras. The image folder contained the three images (left, right, and center) for every captured frame.

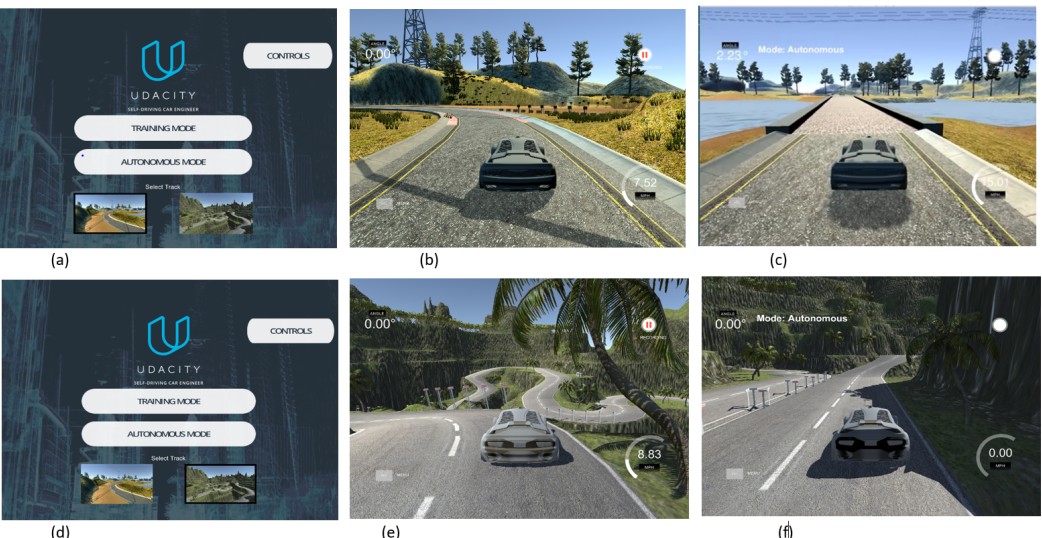

**Figure 5.** The Lack track has three scenarios: (**a**) Lack track environment; (**b**) data recording screen; (**c**) autonomous mode. The Mountain track has three scenarios: (**d**) Mountain track environment; (**e**) data recording screen; (**f**) autonomous mode.

The autonomous mode could be used to test the proposed model to see if it could auto-drive on both tracks while using decision making to keep the autonomous vehicle in the lane without human interaction and while driving safely, as shown in Figure 5c,f. The speed was 30 km/h during the test of the proposed model in the autonomous mode for both tracks. Figure 6 shows the real dataset of the Lincoln self-driving car samples in different scenarios in the daytime.

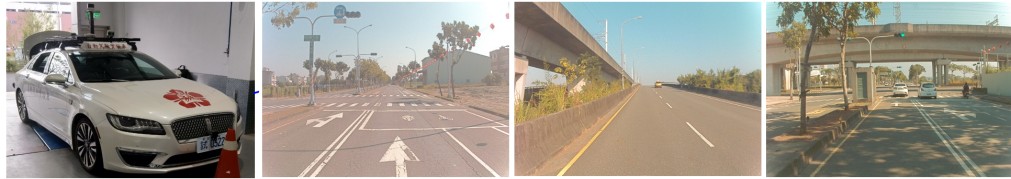

**Figure 6.** The real dataset of the Lincoln self-driving car.

### 4.2. Analysis of the System

Our proposed VGG-GRU framework controlled the predicted steering angle in terms of LT, RT, and LK. Evidently, our proposed framework achieved better performance in terms of the MSE, RMSE, number of parameters, runtime, and total computational time compared with the results of other related works. The batch size and dropout layers were used for overfitting. Adam was selected as the optimizer, and the rectified linear unit (ReLU) was used as the activation function. Moreover, the experimental results of the trained system's loss (MAE), RMSE, and the number of parameters of the simulation for both tracks and the real dataset are shown in Table 3. The proposed model had mean square error (MSE) losses of 0.0230, 0.0520, and 0.0936, respectively. In addition, the RSME values were

0.154, 0.300, and 0.3120 respectively. The proposed model achieved better performance in predicting the steering angle (LT and RT) and speed.

**Table 3.** Performance analysis of the proposed framework in terms of MSE, RMSE, and number of parameters.

| Data | Direction | MSE | RMSE | Parameters |
|------|-----------|-----|------|------------|
| Simulator Lack | $LT_s$ | 0.0230 | 0.154 | 14,927,759 |
| | $RT_s$ | 0.0230 | 0.154 | 149,27,759 |
| | Speed | 0.0233 | 0.0454 | 14,927,759 |
| Simulator Mountain | $LT_s$ | 0.0520 | 0.300 | 15,183,909 |
| | $RT_s$ | 0.0525 | 0.331 | 15,183,909 |
| | Speed | 0.0885 | 0.0400 | 15,183,909 |
| Real data | $LT_s$ | 0.0936 | 0.3120 | 14,929,229 |
| | $RT_s$ | 0.0931 | 0.3038 | 14,929,229 |
| | Speed | 0.0932 | 0.3039 | 14,929,229 |

Herein, the running time was analyzed to calculate two terms: the inference times for a single epoch and the overall proposed architecture. The proposed framework gave a better performance in terms of runtime (Table 4) for a single epoch; the values were 3–4 and 3 ms for both datasets in the simulator and the real dataset respectively.

**Table 4.** Experimental results for the proposed model in terms of inference times and the total time for model execution.

| Data | Direction | Inference Times | Parameters |
|------|-----------|-----------------|------------|
| Lack | $LT_s$ | 3 ms | 14,927,759 |
| | $RT_s$ | 3 ms | 14,927,759 |
| | Speed | 4 ms | 14,927,759 |
| Mountain | $LT_s$ | 3 ms | 15,183,909 |
| | $RT_s$ | 4 ms | 15,183,909 |
| | Speed | 3 ms | 15,183,909 |
| Real-data | $LT_s$ | 3 ms | 14,929,229 |
| | $RT_s$ | 3 ms | 14,929,229 |
| | Speed | 3 ms | 14,929,229 |

For the framework without the proposed VGG-GRU, the performance in terms of MSE, RMSE, the number of parameters, and computation time is shown in Table 5; for a single epoch, the times taken were 3–4 and 3 ms for the simulated and real datasets, respectively.

**Table 5.** Experimental results without the proposed model in terms of MSE, RMSE, parameters, and inference time for model execution.

| Data | Model | MSE | RMSE | Parameters | Inference Time |
|------|-------|-----|------|------------|----------------|
| Lack | VGG16 | 0.1068 | 0.8180 | 14,925,159 | 11 ms |
| Mountain | VGG16 | 0.1211 | 0.7356 | 14,925,159 | 11 ms |
| Real-data | VGG16 | 0.2479 | 0.6791 | 15,802,541 | 10 ms |

Figure 7 summarizes the performance of the proposed model in terms of the MSE (loss). The results show that the proposed framework performed better on both tracks.

The main goal of this paper is the achievement of better results in lane following on road paths.

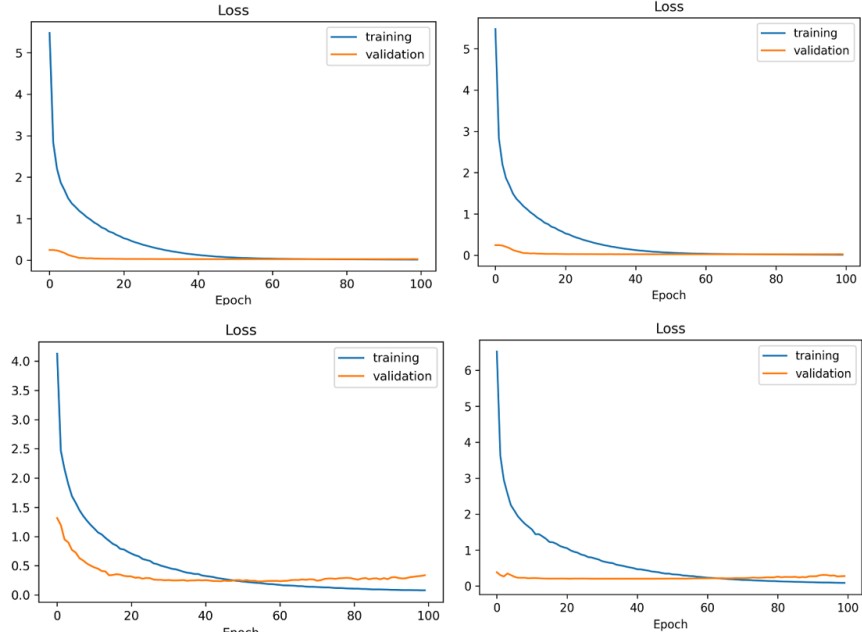

**Figure 7.** Loss metrics: (**Top left**) steering angle and (**Top right**) speed. Loss metrics: (**Bottom right**) steering angle and (**Bottom left**) speed.

Figure 8 shows the proposed model's MSE (loss) performance on the real data. The results show that the proposed framework performed better with the Lincoln self-driving car. The main goal of this paper is the achievement of better results in lane following on road paths.

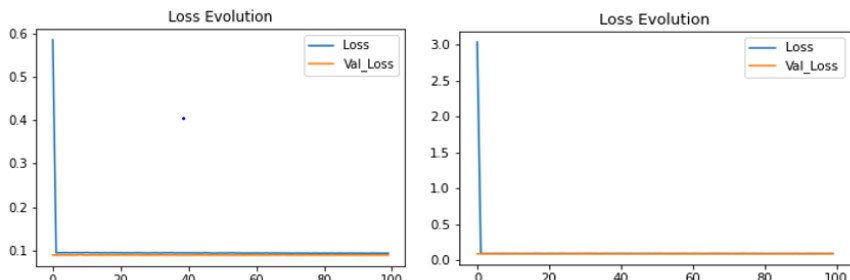

**Figure 8.** Loss metrics: (**left**) steering angle and (**right**) speed on the real dataset.

Table 6 compares the proposed VGG-GRU framework with related models for controlling the steering angle by using our dataset. Evidently, the proposed framework had better performance in terms of MSE, RMSE, the number of parameters, and inference time compared with models from other studies. Table 6 shows that our proposed model gave a better performance compared to the performance of models from previous studies.

**Table 6.** Comparison of the proposed framework with existing frameworks in terms of data, RMSE, inference time, and MSE.

| Ref. | Data/Control | RMSE | Inference Time | MSE |
|:---:|:---:|:---:|:---:|:---:|
| [32] | Udacity | 0.4690 | 8ms | 0.01597 |
| [27] | Udacity | 0.3597 | 6 ms | 0.1113 |
| Our $LT_s$ | | 0.0154 | 3 ms | 0.0230 |
| Our $RT_s$ | Udacity Lack | 0.0154 | 3 ms | 0.0230 |
| Speed | | 0.0454 | 4 ms | 0.0233 |
| Our $LT_s$ | | 0.300 | 3 ms | 0.0520 |
| Our $RT_s$ | U-Mountain | 0.331 | 4 ms | 0.0520 |
| Speed | | 0.0400 | 3 ms | 0.0520 |
| Our $LT_s$ | | 0.3120 | 3 ms | 0.0936 |
| Our $RT_s$ | Real data | 0.3038 | 3 ms | 0.0931 |
| Speed | | 0.3039 | 3 ms | 0.0932 |

*4.3. Embedded System*

Using an implementation on the NVIDIA Jetson 4 GB embedded platform, we showed that our proposed framework can achieve the best performance in terms of computation time when using the behavioral characteristics of a vehicle, such as LT and RT of the steering angle. Table 7 shows the overall performance of the proposed framework when used with an embedded system (Jetson Nano 4 GB) and the GPU, indicating an excellent real-time prediction of the steering angle and speed.

**Table 7.** Comparison of the performances of our proposed framework when used with an embedded system and the GPU in terms of the runtime for a single epoch and accuracy.

| Simulator | Direction | Accuracy | GPU | Jetson Nano | Parameters |
|:---:|:---:|:---:|:---:|:---:|:---:|
| Lack | $LT_s$ | 90% | 25 s | 46 s | 14,927,759 |
| | $RT_s$ | 89% | 25 s | 46 s | 14,927,759 |
| | Speed | 90% | 24 s | 46 s | 14,927,759 |
| Mountain | $LT_s$ | 89% | 23 s | 45 s | 15,183,909 |
| | $RT_s$ | 89% | 23 s | 46 s | 15,183,909 |
| | Speed | 88% | 23 s | 46 s | 15,183,909 |

We estimated the execution time for a training and prediction for a single epoch with the embedded platform. The power consumption of the Jetson Nano is less than that of a GPU when using the same model.

## 5. Conclusions

Autonomous self-driving vehicle technology ensures safe driving and automatic control of the motion of a vehicle. The proposed approach is called the VGG–gated recurrent unit (VGG-GRU). It is based on an understanding of the lane-following planning and control of the steering angle with the speed to continuously keep an autonomous car in the middle of a path during simulated driving situations and real scenarios. In the proposed approach, we utilized a widely adopted deep-learning-network-based architecture, which integrates the VGG-16 and the gated recurrent unit (GRU) for lane following on the road. The experimental results show that the proposed framework predicts steering angles in different directions excellently. The proposed approach (VGG-GRU) achieved higher MSEs of 0.0230 and 0.0932 and better inference times of 3–4 and 3 ms for both the real and simulated scenarios. We also implemented our proposed framework on an embedded system (Jetson Nano 4 GB) and compared its computation time with that when the framework was implemented on a GPU. The results showed that the embedded system took almost 45–46 s to execute a single epoch in order to predict the steering angle.



**Author Contributions:** A.K.: Conceptualization, methodology, writing—original draft preparation; C.-Y.L.: methodology, review, editing, and funding; C.-S.Y.: review and funding. All authors have read and agreed to the published version of the manuscript.

**Funding:** The work was supported in part by the Ministry of Science and Technology of Taiwan under grant numbers MOST 110-2218-E-006-026, 109-2221-E-150-004-MY3 and 111-3116-F-006-005-.

**Institutional Review Board Statement:** Not applicable.

**Informed Consent Statement:** Not applicable.

**Data Availability Statement:** The data analyzed in this study are available from the corresponding authors upon reasonable request.

**Conflicts of Interest:** The authors declare that they have no conflict of interest.

**Sample Availability:** Not applicable.

## Abbreviations

The following abbreviations are used in this manuscript:

| | |
|---|---|
| CNN | Convolutional neural network |
| GRU | Gated recurrent unit |
| AVs | Autonomous vehicle |
| LSTM | Long short-term memory |
| RNN | Recurrent neural network |
| FC | Fully connected |
| LT | Left turn |
| RT | Right turn |
| LK | Lane keeping |
| ReLU | Rectified linear unit |
| MSE | Mean square error |
| RMSE | Root mean square root |
| RNN | Recurrent neural network |
| DNN | Deep neural network |
| HLCIUM | Human-Like Lane-Changing Intention Understanding Model |

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
