# Peer review of "Deep-Learning-Based Network for Lane Following in Autonomous Vehicles"

_electronics, doi:10.3390/electronics11193084_

Round 1
Reviewer 1 Report
Authors are requested to strengthen the experimental investigations of the study. State -of-the the art works with respect to the focused data set and comparison would be resulted.
Reviewer 2 Report
The proposed method uses VGG16 and GRU to perform lane following while autonomous navigation. My comments are as follows:
1. The key contribution is not clear. clearly, the VGG16 and GRU is not the contribution of this paper. How the integration algorithm works and improves the lane following the performance.
2. Results must show the performance with and without the main contribution component.
3. Figure 3 contains 3 parts; a, b, and c. Please add the details about the parts.
4. If possible, please show the performance on real data. (non simulation dataset)
Reviewer 3 Report
The novelty of this work is limited as there are is a lot of work on autonomous driving and lane-following solutions.
The results are based on a simulator rather than images from actual roads. This poses a difficulty in understanding how the system will actually perform. Some experiments on actual images should be conducted and added.
The results indicate that the number of parameters varies with road type. It is not clear how this will work in practice. Is switching of the model expected during operation? There are are also further variety of roads, how are these handled?
There is a Theorem 1, however this is not a theorem.
Reviewer 4 Report
The major issue of this paper is that the novelty is unclear. The authors do not explain how they come up with the proposed network, and they do not describe why they design the network like this. Why a VGG-16 + GRU can work for the lane following case? Is this a random decision or it has detailed derivation steps? Details are needed to support this paper's novelty.
I would like the authors also can address the following questions:
1. Explain the novelty with more details.
2. The related work should be improved. The section 2 describes a lot of unrelated papers. What I hope to see is that the authors can show how other academia papers or industry companies do the lane-following in autonomous vehicle. As lane following is not a new technique, there should be many related published papers. The authors should think about how to make their work different from others.
3. The methodology sections needs a subsection or at least a paragraph to explain how to use the proposed network in reality. For instance, it needs to be trained first and then do the inference on vehicles. Please explain how you deploy this method on vehicles.
4. In Section 4, only training has been analyzed. But I think inference is more essential for this use case, because lane following is performed during the car is driving. And, in table 4, authors need to compare not only training time, but also inference time which is more important.
5. The introduction should be re-written. Again, authors should fully motivate why they design this network in introduction. Also, proof read is needed. For example, in paragraph 2, "... consist of three-part, such as input, 36 deep network-based framework, and decision-making control task." Not all autonomous vehicles are using neural network based methods. Also, a typical autonomous vehicle work flow should be localization, perception and planning. Input is not a module of the autonomous driving framework itself. Another example, "... The deep network- 37 based model contain sensors to perceive the environment as the input, e.g., camera.." Network model does not contain any hardware things. The sensors belong to the vehicle platform. Such errors should definitely be avoided.
Round 2
Reviewer 1 Report
The authors have responded the corrections which have been suggested in the earlier review. The Manuscript is in the current form may be suitable for publication.
Reviewer 3 Report
The authors have improved the paper and included experiments with real data.
Some minor corrections are needed:
Page 5 - "Initially, different networks were considered, such as the VGG16 and the GRU." -> "Initially, different networks were considered, including VGG16 and GRU."
Page 6 - "The original image format is in BGR, so we convert it into BGR and we are not again processed into YUV format." - This is not clear and needs rewording.
Page 11 - "Fig 8 The real data has summarized the proposed model’s MSE (loss) performance." -> "Fig 8 shows the proposed model’s MSE (loss) performance on the real data."
In Figure 8 there are no bottom figures, so change (Top left) -> (left) and (Top right) -> (right).
Reviewer 4 Report
The authors have answered five questioned asked by me in the first round but I do not think my questions are well addressed. I will comment each response in the same order here.
1. The novelty is still not clear. In the author's response, they just simply copied these sentences "Herein, we introduce a deep learning-based method for lane-following ...... " from the third paragraph to the second one. Also, the changes in paragraph 3 just repeated the same materials with a little bit different descriptions. They still did not claim why they propose this architecture and why this work, moreover, why this is better than others' methods.
2. What I want to see is that the authors should add some more related references and compare their design with others' and claim what the difference and why this method is better, rather than simply listing more reference in the end, which makes no sense to readers.
3. I was asking that the authors should explain how to use the trained model in the onboarding system on AVs, but they did not even explain a word on this. How do you let your model execute on AVs to perform lane following? Training is the first step to get a model then the model can be deployed on AVs. I highly doubt the authors do not have conducted real experiments.
4. I asked the authors to include inference measurement in Table 4 but the Table 4 does not have the related info updated.
5. Same issue with comment 1.
Round 3
Reviewer 4 Report
accepted